# Typing assumptions improve identification
# in causal discovery

**Philippe Brouillard**                                   philippe.brouillard@umontreal.ca
*Mila, Université de Montréal*
*ServiceNow Research*

**Perouz Taslakian**                                      p.taslakian@samsung.com
*Samsung AI Center Montreal*

**Alexandre Lacoste**                                     alexandre.lacoste@servicenow.com
*ServiceNow Research*

**Sébastien Lachapelle**                                  sebastien.lachapelle@umontreal.ca
*Mila, Université de Montréal*

**Alexandre Drouin**                                      alexandre.drouin@servicenow.com
*ServiceNow Research*

**Editors:** Bernhard Schölkopf, Caroline Uhler and Kun Zhang

## Abstract

Causal discovery from observational data is a challenging task that can only be solved up to a set of equivalent solutions, called an equivalence class. Such classes, which are often large in size, encode uncertainties about the orientation of some edges in the causal graph. In this work, we propose a new set of assumptions that constrain possible causal relationships based on the nature of variables, thus circumscribing the equivalence class. Namely, we introduce *typed directed acyclic graphs*, in which variable types are used to determine the validity of causal relationships. We demonstrate, both theoretically and empirically, that the proposed assumptions can result in significant gains in the identification of the causal graph. We also propose causal discovery algorithms that make use of these assumptions and demonstrate their benefits on simulated and pseudo-real data.

**Keywords:** causal discovery, structure learning, identification, background knowledge

## 1. Introduction

Can the temperature of a city alter its altitude (Peters et al., 2017)? Can a light bulb change the state of a switch? Can the brakes of a car be activated by their indicator light (de Haan et al., 2019)? Chances are, you did not need to think very hard to answer these questions, since you intuitively understand the implausibility of causal relationships between certain *types of entities*. This form of prior knowledge has been shown to play a key role in causal reasoning (Griffiths et al., 2011; Schulz and Gopnik, 2004; Gopnik and Sobel, 2000). In fact, in the absence of evidence (e.g., data), humans tend to reason inductively and use domain knowledge to generalize known causal relationships to new, similar, entities (Kemp et al., 2010).

Nonetheless, the elucidation of causal relationships often goes beyond human intuition. The abundance of large-scale scientific endeavors to understand the causes of diseases (1KGP, 2010) or natural phenomena (Runge et al., 2019) are good examples. In such cases, computational

methods for *causal discovery* may help reveal causal relationships based on patterns of association in data (see Heinze-Deml et al. (2018) for a review). The most common setting consists of representing causal relationships as a directed acyclic graph where vertices correspond to variables of interest and edges indicate causal relationships. Additional assumptions, like the *faithfulness* condition, are then made to enable reasoning about graph structures based on conditional independences in the data. While these enable data-driven causal discovery, the underlying causal graph can only be identified up to its *Markov equivalence class* (Peters et al., 2017), which can often be very large (He et al., 2015) thus leaving many edges unoriented.

Inspired by how humans use types to reason about causal relationships, this work explores how prior knowledge about the nature of the variables can help reduce the size of such equivalence classes. Building on the theoretical foundations of causal discovery in directed acyclic graphs, we propose a new theoretical framework for the case where *variables are labeled by a type*. Such types can be attributed based on prior knowledge, e.g., via a domain expert. We then make assumptions on how types can interact with each other, which constrains the space of possible graphs and leads to reduced equivalence classes. We show, both theoretically and empirically, that when such assumptions hold in the data, significant gains in the identification of causal relationships can be made.

**Contributions:**

- We propose a new theoretical framework for causal discovery where possible causal relationships are constrained based on the type of variables (Section 4).

- We prove theoretical results that guarantee the orientation of all inter-type edges and, in certain conditions, the convergence of the equivalence class to a singleton (identification), when the number of vertices tends to infinity and the number of types is fixed (Section 5).

- We present simple algorithms to incorporate our type-based assumptions in causal discovery, along with theoretical results that guarantee their consistency (Section 6).

- We present an empirical study that illustrates the benefits of our proposed algorithms over a baseline that does not consider variable types (Section 7).

## 2. Problem formulation

**Causal graphical models.** In this work, we adopt the framework of causal graphical models (CGM) (Peters et al., 2017). Let $X = (X_1, \ldots, X_d)$ be a random vector with distribution $P_X$. Let $G = (V, E)$ be a directed acyclic graph (DAG) with vertices $V = \{v_1, \ldots, v_d\}$. Each vertex $v_i \in V$ is associated to variable $X_i$ and a directed edge $(v_i, v_j) \in E$ represents a direct causal relationship from $X_i$ to $X_j$. We assume that $P_X$ can be factorized according to $G$, that is,

$$p(x_1, \ldots, x_d) = \prod_{i=1}^{d} p(x_i \mid \mathrm{pa}_i^G),$$

where $\mathrm{pa}_i^G$ denotes the parents of $X_i$ in $G$.[1] From this graph, it is possible to estimate quantities of causal nature (e.g., via do-calculus (Pearl, 1995)). However, in many situations, the structure of $G$ is unknown and must be inferred from data.

---

1. This is a slight abuse of language. Here, we mean the parents of vertex $v_i$ in $G$.

**Causal discovery.**   The task of causal discovery consists of learning the structure of $G$ based on observations from $P_X$. Some assumptions are required to make this possible. By adopting the CGM framework, we assume: (i) *causal sufficiency*, which states there is no unobserved variable that causes more than one variable in $X$ and (ii) the *causal Markov property*, which states that $X_i \perp\!\!\!\perp_G X_j \mid Z \implies X_i \perp\!\!\!\perp_{P_X} X_j \mid Z$, where $Z$ is a set composed of variables in $X$, $X_i \perp\!\!\!\perp_G X_j \mid Z$ indicates that $X_i$ and $X_j$ are *d*-separated by $Z$ in $G$, and $X_i \perp\!\!\!\perp_{P_X} X_j \mid Z$ indicates that $X_i$ and $X_j$ are independent conditioned on $Z$. Additionally, we assume (iii) *faithfulness*, which states that $X_i \perp\!\!\!\perp_{P_X} X_j \mid Z \implies X_i \perp\!\!\!\perp_G X_j \mid Z$. Hence, conditional independences in the data can be used to learn about the structure of $G$.

**Equivalence classes.**   Even with these assumptions, $G$ can only be recovered up to a *Markov equivalence class* (MEC) (Peters et al., 2017), which is the set of all the DAGs that encode exactly the same conditional independences as $G$. The MEC is often characterized graphically using an *essential graph* or *Completed Partially Directed Acyclic Graph* (CPDAG), which corresponds to the union of all Markov equivalent DAGs (Andersson et al., 1997). While two DAGs are Markov equivalent if and only if they have the same skeleton and *v-structures* (also called *immoralities*) (Verma and Pearl, 1990), the CPDAG can contain other oriented edges, resulting from constraints such as not creating cycles or additional v-structures.[2] In some cases, e.g., for sparse graphs, the size of the MEC can be huge (He and Yu, 2016; He et al., 2015), significantly limiting inference about the direction of edges in $G$. Hence, it is a problem of key importance to find new realistic assumptions to shrink the equivalence class.

There have been a wealth of approaches to alleviate this problem. For instance, some have made progress by including data collected under intervention (Hauser and Bühlmann, 2012), making assumptions about the functional form of causal relationships (Peters et al., 2014; Shimizu et al., 2006) or including background knowledge on the direction of edges (Meek, 1995). In this work, we propose an alternative approach, based on background knowledge, where types are attributed to variables and the interaction between types is constrained.

## 3. Related work

The inclusion of background knowledge in causal discovery aims to reduce the size of the solution space by adding or ruling out causal relationships based on expert knowledge. Several forms of background knowledge have been proposed, which place various levels of burden on the expert. Below, we outline those most relevant to our work (see Constantinou et al. (2021) for a review).

**Hard background knowledge.**   This type of background knowledge is "hard" in the sense that it *must be respected* in the inferred graph structures. Previous works have considered: sets of forbidden and known edges (Meek, 1995), a known ordering of the variables (Cooper and Herskovits, 1992), partial orderings of the variables (Andrews, 2020; Scheines et al., 1998), and ancestral constraints (Li and Beek, 2018; Chen et al., 2016). Among these, partial orderings (or *tiered background knowledge*) are the most similar to our contribution. In this setting, it is assumed that an expert partitions the variables into sets called *tiers*, and orders the tiers such that variables in a later tier cannot cause variables in an earlier tier. In contrast, while we require an expert to partition variables into sets (by type), we do not assume that an ordering is known *a priori* (see Appendix E.1 for examples).

---

2. For our terminology related to graphs, we refer the reader to the Appendix A of Andersson et al. (1997).

Figure 1: **(a)** Representation of t-edges orientations, where colors represent the different types $t_a$, $t_b$, and $t_c$. Representation of a t-DAG that is consistent and follows the orientation of the t-edges in (a). **(c)** Representation of a t-DAG that is not consistent: the red dotted edge $v_{c_2} \to v_{b_1}$ is not consistent with $v_{b_1} \to v_{c_1}$ (Definition 3).

**Soft background knowledge.** A setting similar to ours, where the type of each variable dictates its possible causal relationships, is presented by Mansinghka et al. (2012). They propose a Bayesian method to use this prior knowledge in causal discovery. Their work shows the benefits of such priors, but does not investigate this space of graphs and their properties w.r.t. to structure identifiability.

**Grouping variables.** Parviainen and Kaski (2017) explore a setting similar to ours, where variables representing different "views" on the same entity are aggregated into groups. The authors address the problem of learning causal relationships between groups of variables, which they represent as *group DAGs*. Our work is conceptually different. First, variables of a given type could correspond to different entities that are similar, rather than multiple views on a common entity. Second, our focus is different: their goal is to recover a group DAG, while ours is to make assumptions that facilitate the identification of the causal graph in the variable space. Note, however, that their *strong group causality* assumption leads to graphs that are a subset of the consistent t-DAGs that we will present.

Interestingly, several recent works applying causal discovery to real-world problems rely on expert knowledge that is compatible with our proposed framework. For example, in their work on Alzheimer's disease, Shen et al. (2020) claim that "edges from biomarkers or diagnosis to demographic variables are prohibited" and that "edges among demographic variables are prohibited", clearly reasoning about relationships between types of variables. Similarly, the work of Flores et al. (2011) outlines an application of tiered background knowledge in a medical case study. Converting this setting to ours simply involves considering each tier as a variable type. Hence, the typing assumptions that we propose in this work, and the associated theoretical results, constitute a way of incorporating expert knowledge that is applicable in practice.

## 4. Typed directed acyclic graphs

Our work builds on two fundamental structures: *typed* directed acyclic graphs (t-DAG), which are essentially DAGs with typed vertices; and *t-edges*, which are sets of edges relating vertices of distinct types. Formal definitions follow.

**Definition 1 (t-DAG)** *A t-DAG $D_T$ with $k$ types is a DAG $D := (V, E)$ augmented with a mapping $T : V \to \mathcal{T}$ such that the type of $v_i \in V$ is $T(v_i) = t_j \in \mathcal{T}$, where $|\mathcal{T}| = k$.*

**Definition 2 (t-edge)** *A t-edge $E(t_i, t_j)$ is the set of edges that goes from a vertex of type $t_i$ to a vertex of type $t_j$. More formally, $E(t_i, t_j) = \{(v_k, v_l) \in E \mid T(v_k) = t_i, T(v_l) = t_j\}$ for any pair of types $t_i, t_j \in \mathcal{T}$, s.t., $t_i \neq t_j$.*

For example, the graphs illustrated in Fig. 1 (b) and (c) are t-DAGs where colors represent types and the set $E(t_a, t_c) = \{(v_{a_1}, v_{c_1}), (v_{a_1}, v_{c_2})\}$ is a t-edge between types $t_a$ and $t_c$.[3]

### 4.1. Assumptions on type interactions

We now introduce a new assumption: *type consistency*, which constrains the possible causal relationships that may arise between typed variables. Put simply, this assumption states that causal relationships between two types of variables can only arise in one common direction.[4]

**Definition 3 (Consistent t-DAG)** *A consistent t-DAG is a t-DAG where, for every pair of distinct types $t_i, t_j$, if t-edge $E(t_i, t_j) \neq \emptyset$ then we have that $E(t_j, t_i) = \emptyset$. We refer to this structural constraint as type consistency. For conciseness, $t_i \xrightarrow{t} t_j$ denotes $E(t_i, t_j) \neq \emptyset$.*

In Fig. 1 (b), we present an example of a consistent t-DAG. In contrast, the t-DAG shown in Fig. 1 (c) is not consistent: the t-edge $E(t_c, t_b)$ (purple to white) contains the edge $(v_{c_2}, v_{b_1})$, while the reverse t-edge, $E(t_b, t_c)$, is not empty since it contains $(v_{b_1}, v_{c_1})$. Notice how the orientation of all t-edges (Fig. 1 (a)) fully determines the orientation of edges between variables of distinct types in a consistent t-DAG.

Note that alternative assumptions could have been considered. For instance, we could have assumed that t-edges form a DAG (i.e., the types have a partial ordering). However, the assumptions considered here are less restrictive and, as we demonstrate later, lead to interesting results.

### 4.2. Equivalence classes for consistent t-DAGs

We define the equivalence classes MEC and t-MEC as the set of DAGs and the set of consistent t-DAGs that are Markov equivalent, respectively.

**Definition 4 (MEC)** *The MEC of a t-DAG $D_T$ is $M(D_T) := \{D' \mid D' \sim D_T\}$ where "$\sim$" denotes Markov equivalence.*

**Definition 5 (t-MEC)** *The t-MEC of a consistent t-DAG $D_T$ is $M_T(D_T) := \{D'_T \mid D'_T \overset{t}{\sim} D_T\}$ where "$\overset{t}{\sim}$" denotes Markov equivalence limited to consistent t-DAGs with the same type mapping $T$.*

---

3. To keep the figures simple and readable, throughout the paper we label the vertices of the t-DAGs with the subscripts of the variables (or types) they represent. For example, vertex $a_i$ refers to variable $v_{a_i}$ and vertex $a$ refers to type $t_a$.

4. See Appendix E.2 for a discussion of variations and relaxations of this typing assumption.

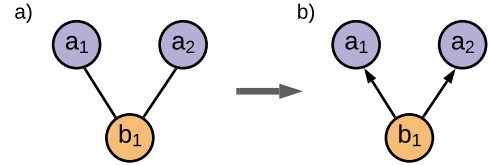

Figure 2: **(a)** The two-type fork structure. In this illustration, the vertices $v_{a_1}$ and $v_{a_2}$ are of type $t_a$ (purple) and $v_{b_1}$ is of type $t_b$ (orange). **(b)** Orientation rule: if this structure is encountered in an essential graph, it must be oriented in the t-essential graph.

To represent an equivalence class, we can use an *essential graph*, which corresponds to the union of equivalent DAGs. The union over graphs is defined as the union of their vertices and edges: $G_1 \cup G_2 := (V_1 \cup V_2, E_1 \cup E_2)$. Also, if $(v_i, v_j), (v_j, v_i) \in E_1 \cup E_2$, then the edge is considered to be undirected.

**Definition 6 (Essential graph)** *The essential graph $D^*$ associated to the consistent t-DAG $D_T$ is*

$$D^* := \bigcup_{D \in M(D_T)} D.$$

**Definition 7 (t-Essential graph)** *The t-essential graph $D_T^*$ associated to the consistent t-DAG $D_T$ is*

$$D_T^* := \bigcup_{D \in M_T(D_T)} D.$$

### 4.3. t-Essential graph properties and size of t-MEC

We consider some statements that can directly be made about t-essential graphs and the size of t-MEC with respect to their non-typed counterparts. Proofs for the propositions can be found in Appendix A. First note that for t-DAGs with $k$ types and $d$ vertices, in the limit cases where each variable belongs to a distinct type ($k = d$) or all variables belong to a single type ($k = 1$), type consistency does not impose structural constraints on t-DAGs, i.e., any t-DAG is type-consistent and the t-essential graph is identical to the essential graph.

However, in general, the t-essential graph is a version of the essential graph with more oriented edges, thanks to the type consistency assumption:

**Proposition 8** *Let $D_T^*$ and $D^*$ be, respectively, the t-essential and essential graphs of an arbitrary consistent t-DAG $D_T$. Then, $D_T \subseteq D_T^* \subseteq D^*$.*

Indeed, type consistency synchronizes the orientation of some edges, resulting in a reduced set of possible orientations. Some structural properties of the graph may also force the orientation of edges in the t-essential graph. For instance, akin to v-structures in essential graphs, *two-type forks* (see Fig. 2) must be oriented in t-essential graphs.

**Proposition 9** *If a consistent t-DAG $D_T$ contains vertices $v_{a_1}, v_{a_2}, v_{b_1}$ with types $T(v_{a_1}) = T(v_{a_2}) = t_a$, $T(v_{b_1}) = t_b$ and $t_a \neq t_b$, with edges $v_{a_1} \leftarrow v_{b_1} \rightarrow v_{a_2}$ ($v_{a_1}, v_{a_2}$ not adjacent), then*

the t-edge $t_b \xrightarrow{t} t_a$ is directed in the t-essential graph, i.e., the direction of causation between types $t_b$ and $t_a$ is known.

To see this, note that, under type consistency, there are only two possible orientations: $v_{a_1} \to v_{b_1} \leftarrow v_{a_2}$ and $v_{a_1} \leftarrow v_{b_1} \to v_{a_2}$. The first is a v-structure and, thus, must be oriented in the essential graph. If it is not, there is only one possible alternative orientation. Therefore, such edges are always oriented in the t-essential graph.

Furthermore, we can upper bound the size of the t-MEC based on the number of undirected edges in the t-essential graph, as stated in the following proposition.

**Proposition 10 (Upper bound on the size of the t-MEC)** *For any consistent t-DAG $D_t$, we have $|M_T(D_T)| \le 2^u \prod_{t_i \in \mathcal{T}} 2^{u_{t_i}}$, where $u$ and $u_{t_i}$ are respectively the number of undirected t-edges and the number of undirected edges between variables of type $t_i$ (intra-type edges) in the t-essential graph of $D_t$.*

From this bound, we can also directly conclude that if the t-essential graph contains no undirected edges, then $|M_T(D_T)| = 1$. In other words, $D_T$ is identified.

## 5. Identification for random graphs

In this section, we explore the benefits of variable typing in causal graph identification through the study of a class of graphs generated at random based on a process inspired by the Erdős-Rényi random graph model (Erdős and Rényi, 1959).

Assume we are given a set of $k$ types $t_1, \ldots, t_k$, probabilities $p_1, \ldots, p_k \in (0,1)^k$ of observing each type s.t. $\sum p_i = 1$, and a *type interaction matrix* $A \in [0,1]^{k \times k}$ where each cell $(i,j)$ is the probability $p_{ij}$ that a variable of type $t_i$ is a direct cause of a variable of type $t_j$. As per Definition 3 (type consistency), we impose that $\forall i \ne j$, if $p_{ij} > 0$, then $p_{ji} = 0$.

**Definition 11 (Random sequence of growing t-DAG)** *We define a random sequence of t-DAGs $(D_{T^n}^n)_{n=0}^{\infty}$ with $D_{T^n}^n = (V^n, E^n)$ and $|V^n| = n$, such that $D_{T^0}^0 = (\emptyset, \emptyset)$. Each new t-DAG $D_{T^n}^n$ in the sequence is obtained from $D_{T^{n-1}}^{n-1}$ as follows: Create a new vertex $v_n$ and sample its type $t$ from a categorical distribution with probabilities $p_1, ..., p_k$. Let $V^n = V^{n-1} \bigcup \{v_n\}$. Let $T^n(v_n) = t$ and $T^n(v_j) = T^{n-1}(v_j), \forall v_j \in V^{n-1}$; To obtain $E^n$, for every vertex $v_i \in V^{n-1}$, add the edge $(v_i, v_n)$ to $E^{n-1}$ with probability $p_{T^n(v_i), T^n(v_n)}$.*

Our main theorem below states that as we add more vertices to such a growing sequence of random t-DAGs, we eventually discover the orientation of all t-edges. We defer the proof to Appendix B.1.

**Theorem 12** *Let $(D_{T^n}^n)_{n=0}^{\infty}$ be a random sequence of growing t-DAGs as defined in Definition 11, let $U$ be the number of unoriented t-edges, and let $r_{ij} = -\frac{1}{3} \max \left[ \ln(1-p_i), \ln\left(1 - p_j p_{ij}(1-p_{jj})\right) \right]$. For $n \ge 3$, $p_i, p_j \in (0,1)$, and $p_{ij}, p_{jj} \in [0,1]$, we have:*

$$P(U > 0) \le 4 \sum_{i,j \,:\, i \ne j, p_{ij} > 0} e^{-r_{ij}n}.$$

To give an intuition of the proof, recall Proposition 9, which tells us that any two-type fork structure must be oriented in the t-essential graph, thereby orienting the associated t-edge. We thus argue that, as we add more vertices, the probability of observing a two-type fork for arbitrary type pairs converges to 1. This argument relies on the fact that the number $k$ of types remains constant throughout as the random t-DAG grows.

From Theorem 12 we can derive a result for the case where variables of the same type do not interact ($p_{ii} = 0, \forall i$). In this case, the t-MEC collapses to a singleton as the graph grows, resulting in identification of the true t-DAG.

**Corollary 13** *Let $(D_{T^n}^n)_{n=0}^\infty$ be a random sequence of growing t-DAGs as defined in Definition 11 and let $r_{ij} = -\frac{1}{3} \max\left[\ln(1 - p_i), \ln\left(1 - p_j p_{ij}\right)\right]$. For $n \geq 3$, $p_i, p_j \in (0, 1)$, and $p_{ij} \in [0, 1]$, the size of the t-MEC converges to 1 exponentially fast:*

$$P(|M_T(D_{T^n}^n)| > 1) \leq 4 \sum_{i,j \,:\, i \neq j, p_{ij} > 0} e^{-r_{ij} n}.$$

### 5.1. Empirical validation

We conduct an empirical study to validate these theoretical results and further compare the size of the MEC and t-MEC for the case where $p_{ii} > 0$. As such, we consider t-DAGs of various sizes, randomly generated according to the process described in Definition 11. Let $k$ be the number of types in the t-DAG. We attribute uniform probability to each type, i.e., $p_i = 1/k$, $\forall i \in \{1, ..., k\}$. The type interaction matrix $A$ is defined as follows. For each pair of types $(t_i, t_j)$, s.t., $i \neq j$, the direction of the t-edge is sampled randomly with uniform probability and we use a fixed probability of interaction $p_{\text{inter}}$. For example, if the direction $t_i \xrightarrow{t} t_j$ is sampled, then $A_{ij} = p_{\text{inter}}$ and $A_{ji} = 0$. Furthermore, for each type $t_i$, we attribute a fixed probability $p_{\text{intra}}$ for the occurrence of edges between variables of type $t_i$.

Fig. 3 shows results for t-DAGs with $n = \{10 \dots 100\}$ vertices, $k = 10$ types, $p_{\text{inter}} = 0.2$, and various values for $p_{\text{intra}}$ (see Appendix B.2 for additional results). In Fig. 3($a$), we clearly see that as, the number of vertices grows, the number of unoriented t-edges tends to zero irrespective of the value of $p_{\text{intra}}$, supporting the statement of Theorem 12. Furthermore, Fig. 3($b$) clearly shows that for the case where $p_{\text{intra}} = 0$, the size of the t-MEC tends to 1 (identification) as the number of vertices grows, supporting the statement of Corollary 13. In sharp contrast, the size of the MEC grows with the number of vertices. Finally, Fig. 3($c$) shows that the size of the t-MEC can be much smaller than that of the MEC, even when the t-DAG contains intra-type edges ($p_{\text{intra}} > 0$), for which we do not have orientation guarantees. Interestingly, this also holds when the t-DAGs are dominated by intra-type edges (e.g., $p_{\text{intra}} = 0.5$).

It remains an open question to formally quantify the size of the t-MEC vs. the MEC when the graphs contain intra-type edges. The results in Fig. 3($c$) suggests that their ratio may be bounded by a quantity that depends on $p_{\text{intra}}$.

## 6. Causal discovery algorithms for t-essential graphs

Causal discovery algorithms, such as the PC algorithm (Spirtes et al., 2000), are typically consistent w.r.t. the MEC. That is, given infinite samples from the observational distribution entailed by a causal graph, they are guaranteed to recover its essential graph. Given that

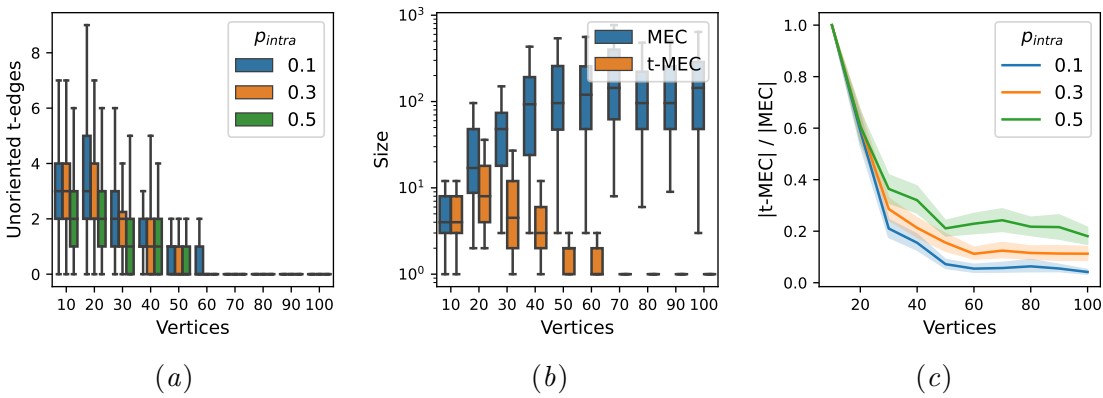

Figure 3: (a) Number of unoriented t-edges w.r.t. the number of vertices (b) Size of the MEC and the t-MEC w.r.t. the number of vertices when $p_{\text{intra}} = 0$ (c) Ratio of the size of t-MEC and the size of MEC w.r.t. the number of vertices.

the t-MEC is often a small subset of the MEC, it is desirable to find algorithms that can consistently recover t-essential graphs. In this section, we present such algorithms.

### 6.1. From essential to t-essential graph

Given an essential graph, one can recover the t-essential graph by enumerating all Markov equivalent consistent t-DAGs and taking their union. We propose a slightly more efficient approach that propagates t-edge orientations based on the Meek (1995) orientation rules:

**Algorithm 1:** t-Propagation$(G, T)$

1. Enforce type consistency: If there exists an oriented edge between any pair of variables with types $t_i, t_j \in \mathcal{T}$ in $G$, assume $t_i \xrightarrow{t} t_j$ and orient all edges between these types.

2. Apply the Meek (1995) orientation rules R1-R4 (see their Section 2.1.2) to propagate the edge orientations derived in Step (1).

3. Repeat from Step (1) until the graph is unchanged.

4. Enumerate all t-DAGs that can be produced by orienting edges in the resulting graph. Reject any inconsistent t-DAGs and take the union of all remaining t-DAGs to obtain the t-essential graph (see Definition 7).

If $G$ is a graph with the same skeleton, v-structures, and type mapping $T$ as the t-essential graph $D_T^*$, then Algorithm 1 is guaranteed to recover the corresponding t-essential graph $M_T(D_T)$ (see Appendix C.1).

Thus, Algorithm 1 can be used in conjunction with any MEC-consistent causal discovery algorithm to obtain one that is t-MEC-consistent. However, one major caveat is that, for finite sample sizes, the output of the MEC-consistent algorithm may violate type consistency and cause an irrecuperable failure of t-Propagation (impossibility of orienting t-edges consistently).

Another limitation of this algorithm is its non-polynomial time complexity due to the enumeration in Step (4). Without it, the algorithm would be *sound*, i.e., it would not orient edges that are unoriented in the t-essential graph, but not *complete*, i.e. some edges that should be oriented in the t-essential graph would remain unoriented. To see this, consider the *two-type fork* illustrated in Fig. 2(a). The essential graph for this t-DAG would be completely undirected since it contains no v-structures. This would result in a case where none of the Meek (1995) rules are applicable and thus, Step (2) would not orient any edges. The algorithm would therefore stop and return a fully undirected graph. However, according to Proposition 9, the two-type fork should have been oriented.

Nevertheless, even with an additional rule to orient such structures in Step (2) (as illustrated in Fig. 2(b)), the algorithm would not be complete without Step (4). In Appendix C.1.2, we show more complex counterexamples (non-local and involving multiple t-edges). It thus remains an open question whether it is possible to design a polynomial-time algorithm to find the t-essential graph, as Meek (1995) and Andersson et al. (1997) did for essential graphs. Note, however, that the non-polynomial time complexity was found to be non-prohibitive in our experiments.

## 6.2. Typed variants of the PC algorithm

The previous algorithm suffices to achieve consistency w.r.t. the t-MEC, but it may fail to produce type-consistent outputs when used with finite sample sizes. Here, we show that it is possible to modify the PC algorithm (Spirtes et al., 2000) to obtain a t-MEC-consistent algorithm that always produces type-consistent outputs.

Recall how the PC algorithm works; it proceeds in three phases: 1) infer the graph's skeleton using conditional independence tests[5], 2) orient all v-structures in the skeleton, 3) apply the Meek (1995) rules to orient as many edges as possible given the edges oriented in Phase (2). To ensure type-consistent outputs, it suffices to modify Phases (2) and (3) to ensure that, when applicable, we orient whole t-edges instead of single edges.

For Phase (3), the modification is simple: replace the usual procedure by t-Propagation (Algorithm 1). For Phase (2), one must deal with errors that may arise when applying conditional independence tests to samples of finite size. Indeed, it is possible to observe edges from the same t-edge involved in both v-structures and two-type forks (see Fig. 2), suggesting multiple orientations when only one is possible. We consider two simple strategies for dealing with such ambiguities that we outline below. These lead to two *Typed-PC (TPC)* algorithms that are both t-MEC-consistent, but differ in their empirical performance, as we show in Section 7. Pseudo-codes and proofs of consistency are given in Appendix C.2.

- **TPC-naive**: Use the first encountered structure (v-structure or two-type fork) to orient each t-edge. Naturally, this naive strategy is error-prone.

- **TPC-majority:** We know that only one orientation is possible for t-edges. Hence, look at each structure that would trigger an orientation (v-structures and two-type forks) and choose the orientation based on the most frequent type of structure.

---

5. The conditional independence test must be chosen based on the nature of the data. We use FIT (Chalupka et al., 2018), since it is non-parametric and applies to both continuous and discrete data.

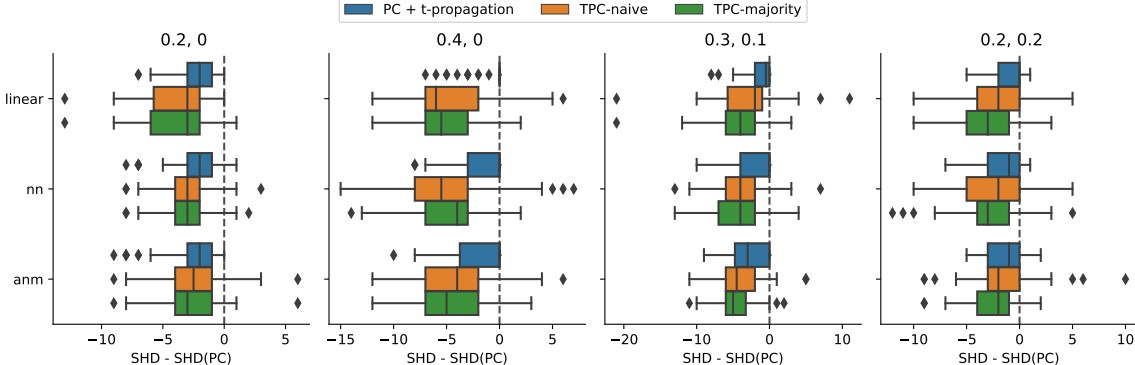

Figure 4: Results on simulated data shown as improvement in SHD w.r.t. the PC baseline (lower is better). The title of each subplot indicates the $(p_{\mathrm{inter}}, p_{\mathrm{intra}})$ configuration.

## 7. Experiments

We conduct causal structure learning experiments[6] to compare the performance of our proposed t-MEC-consistent algorithms with that of a baseline which does not make use of variable types. The t-MEC-consistent algorithms are: TPC-naive and TPC-majority (Section 6.2) and PC (Spirtes et al., 2000) augmented with t-Propagation (Section 6.1). The baseline is the classical PC algorithm. The methods are compared in terms of Structural Hamming Distance (SHD) between their output and the ground-truth t-essential graph (see Appendix D.2 for details). We base the comparison on synthetic and pseudo-real datasets.

**Synthetic data.** We consider graphs randomly generated according to Section 5.1 with 20 vertices and 5 types.[7] The probabilities of connection $p_{\mathrm{inter}}$ and $p_{\mathrm{intra}}$ vary in $\{(0.2, 0), (0.4, 0), (0.3, 0.1), (0.2, 0.2)\}$. The $(0.2, 0)$ configuration results in sparse graphs with no intra-type edges. The others configurations lead to denser graphs with similar densities, but which differ in the abundance of intra-type edges. For each type of graph, we explore multiple parametric forms for the causal relationships: linear, nonlinear additive noise model (ANM) (Bühlmann et al., 2014), and nonlinear non-additive model using neural networks (NN) (Kalainathan et al., 2018). Finally, for each type of graph and parametric form, we generate 50 different consistent t-DAGs and draw 10k samples from their observational distribution.

**Pseudo-real data.** We consider the following Bayesian networks from the Bayesian Network Repository (number of variables in parentheses): `sachs` (11), `child` (20), `insurance` (27), `alarm` (37), `hailfinder` (56), `win95pts` (76). For each network, the conditional probabilities are fitted to real-world data sets, enabling the generation of pseudo-real data. To assign types to variables, we randomly partition their topological ordering into groups of expected size 5 (see Appendix D.1.2). We consider 50 random type assignments and for each, we sample 50k observations from the Bayesian network.

---

6. Implementations of the algorithms and code for the experiments are available at https://github.com/ElementAI/typed-dag.

7. See Appendix D.3 for additional results.

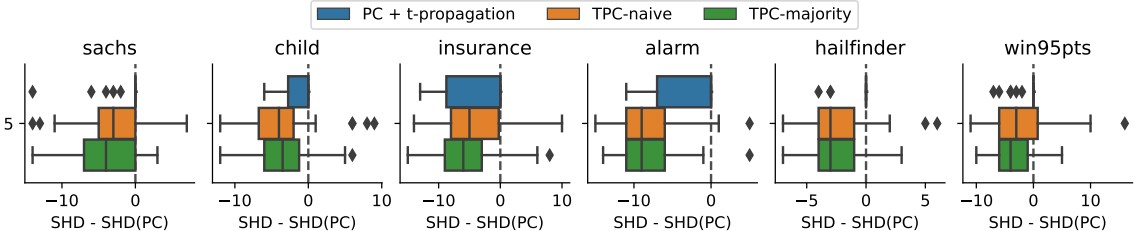

Figure 5: Results on pseudo-real data shown as improvement in SHD w.r.t. the PC baseline (lower is better). The title of each subplot indicates the underlying network.

**Results.** The results reported in Figs. 4 and 5 show the improvement in SHD w.r.t. PC. As expected, all t-MEC-consistent algorithms generally outperform this type-agnostic baseline, achieving lower SHD w.r.t. the t-essential graph. The PC + t-Propagation method sometimes underperforms compared to the TPC algorithms, mainly because it fails to produce a type-consistent output[8] (49% of the time). Finally, as expected, TPC-majority tends to yield better or equal performance compared to TPC-naive, since it is less vulnerable to errors in t-edge orientations that may arise at Phase (2) of the algorithm (see Section 6.2).

## 8. Discussion

In this work, we address an important problem in causal discovery: the fact that it is often impossible to identify the causal graph precisely, due to the size of its Markov equivalence class. This is particularly true for sparse graphs, where the size of the MEC grows super-exponentially with the number of vertices (He et al., 2015). Our theoretical and empirical results clearly demonstrate that there exist conditions under which our variable-typing assumptions greatly shrink the size of the equivalence class. Hence, when such assumptions hold in the data, gains in identification are to be expected. We also propose methods to recover the t-essential graph from data and, using several synthetic and pseudo-real data sets, show that these perform better than their type-agnostic counterparts.

We note that the new assumptions that we introduce can be used in conjunction with other strategies to shrink the size of equivalence classes, such as considering interventions (Hauser and Bühlmann, 2012), hard background knowledge on the presence/absence of edges (Meek, 1995), or functional-form assumptions (Peters et al., 2014; Shimizu et al., 2006).

While this work focuses on causal discovery, it would be interesting to explore the implications of our theoretical framework for causal inference, i.e., the estimation of causal effects. For instance, the small size of t-MECs in comparison to MECs could improve the accuracy of methods that estimate causal effects based on equivalence classes, such as the IDA variant of Perkovic et al. (2017). Also, Anand et al. (2022) recently proposed a method to estimate causal effects in graphs where clusters of variables have unknown relationships. This may prove particularly useful to estimate causal effects based on t-essential graphs with oriented t-edges, but unoriented intra-type edges, since these could be treated as clusters.

---

8. In this case, it simply returns the output of PC.

In addition, we believe that this work may stimulate new advances at the intersection of machine learning and causality (Schölkopf et al., 2021; Schölkopf, 2019). In fact, machine learning algorithms excel at classification, and thus it may be interesting to explore a setting where the variable types are learned based on some variable features. Type assignments could be learned *in parallel with* causal discovery using recent methods for differentiable causal discovery (Brouillard et al., 2020; Zheng et al., 2018). This may further reduce the burden on human experts in cases where types are hard to assign. As an example, consider the task of learning causal models of gene regulatory networks. One could train a model to assign types to genes, based on features of their DNA sequence or their categorization in the gene ontology (Gene Ontology Consortium, 2004), in a process where types are assigned such as to help causal discovery.

Another interesting future direction would be to use our typing assumptions to perform causal discovery on multiple graphs at once, i.e., *multi-task causal discovery*. In fact, assume that we are given data for multiple groups of variables that correspond to disjoint systems (no interactions across groups), but that share similar types. It would be possible to use type consistency (Definition 3) to propagate t-edge orientations across graphs.

In conclusion, our type consistency assumption can result in significant gains in identification that can readily be leveraged by modified versions of common causal discovery algorithms. We believe that assumptions based on types are truly important since, in addition to facilitating causal discovery, they are likely to be a key component of causal reasoning in intelligent agents.

## Acknowledgements

The authors are grateful to Assya Trofimov, David Berger, Hector Palacios, Jean-Philippe Reid, Nicolas Chapados, Pau Rodriguez, Pierre-André Noël, and Sébastien Paquet for helpful comments and suggestions. They also thank the anonymous reviewers for their thoughtful questions and comments. Sébastien Lachapelle is supported by an IVADO Excellence PhD scholarship.

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
