# OpenReview forum: "Typing assumptions improve identification in causal discovery"
_cclear.cc/CLeaR/2022/Conference — CLeaR 2022 Oral_

### Official Review · Reviewer_Nb5F · 2021-11-21

**Confidence:** 4
**Overall Score:** 8

**Main Review:**

Quality: The paper overall is of excellent quality. The technical exposition is up to the standard one would expect for a paper in causal discovery.
Clarity: This paper is very well written. The flow of the paper begins with interesting motivational ideas, and naturally opens into a formulation of the problem
Originality: To my knowledge, I have not seen DAGs approached in this particular manner. Similar techniques have existed (e.g. tiers in causal discovery) but none that restrict the equivalence class.
Significance: I believe this work will be significant to practitioners of causal discovery, since it develops tools that help incorporate additional information often present in applications.
Pros: The paper overall proposes a novel and interesting idea. The authors have done a good job developing and shaping that idea.
Cons: While I do think this work is a step in the right direction, I do wonder how often practitioners will find themselves in a situation where they can a) confidently partition variables into distinct types and b) have no idea how these types relate to each other. In my experience applying these methods, a) and b) tend to move together, since having a basic temporal ordering of variables is quite common in practice.
The authors do address this partially by noting that types could be learned using ML classification methods, although this poses a further question of how exactly types might be identified from data (it is not immediately obvious to me that this identification task is trivial).
Is it possible to incorporate some notion of uncertainty about types? The motivation here is that if I am unsure of how types relate to each other, I am probably also less sure of which variables belong to which type. Perhaps the authors may be able to sketch out a sensitivity analysis strategy that addresses this.
Minor notes:
I would consider using “collider” or “V-structure” in place of “immorality” as a more gender-neutral term.


**Summary:**

This paper develops a new class of graphical models called typed directed acyclic graphs. These tDAGs differ from DAGs in that each node is assigned a single type. Then, assuming that causal relationships between two types of variables arises in one direction, they develop Markov equivalence class theory for t-DAGs. This serves as the foundation for causal discovery, in which the authors modify the PC algorithm to handle types. Finally, the authors consider identification for random tDAGs.

---

> ### Author Response · Authors · 2021-12-02
> **Response to reviewer Nb5F**
>
> Thank you for your comments. The general response (above) contains answers to your questions on the plausibility of an expert being able to assign types without having an intuition of the ordering (Point 3) and an extension where variable types are learned (Point 4). As for your comment on the term *immorality*, we absolutely agree and will replace it by *v-structure*.

---

### Official Review · Reviewer_Q9JM · 2021-11-22

**Confidence:** 3
**Overall Score:** 7

**Main Review:**

In my view this paper checks all of the boxes: it is well written and clear, the idea is a natural one, the results are largely convincing, and one is left wanting to know more. The central idea of the paper is very simple. It is not at all surprising that the kind of inductive bias the authors pinpoint would help in causal inference, simply by reducing the space of compatible causal graphs. But the authors go beyond this basic observation to clarify how exactly it will do so, and also give suggestive case studies in which the proposed algorithms really do appear to lead to gains. I also think the authors have a good argument for why this type of qualitative information is the kind of assumption that a domain expert might realistically be comfortable making.

Most of my questions about extensions and further comparisons are probably already on the authors’ radar. How does this work in the semi-Markovian setting (latent confounders)? What about other causal structure learning algorithms other than PC? And so on.

One point in the discussion that made me a bit nervous was the authors’ discussion of automated ways of discovering types. I could not tell from the discussion exactly what they had in mind here, but I believe it’s important that types in this context are not simply categories: rather, they have a substantive causal flavor. Without some further reason for thinking that categories in a given domain line up nicely with types that would support the relevant kinds of inference, I worry that trying to induce the types from correlational data could lead to systematic faulty inferences. Perhaps the authors could say a bit more about what they had in mind there?

**Summary:**

This paper proposes a method for incorporating types into causal models, with corresponding causal structure learning algorithms for respecting typing assumptions. In short, if t1 and t2 are different types, then at most one type can cause the other. The utility of the method is shown in theory and in some test cases.

---

> ### Author Response · Authors · 2021-12-02
> **Response to reviewer Q9JM**
>
> Thank you for your comments. The general response (above) contains answers to your questions on an extension to the semi-Markovian setting (Point 1), other causal structure learning algorithms (Point 2), and an extension where variable types are learned (Point 4).

---

### Official Review · Reviewer_Dzzm · 2021-11-23

**Confidence:** 3
**Overall Score:** 6

**Main Review:**

The authors show that when a causal graph is assumed to be constituted of vertices belonging to specific types, and that edges are assumed to exist only in one direction between any two classes, the resulting Markov equivalence set for the data distribution in question becomes smaller. The authors' presentation is clear and easy to follow. This can be interesting for the researchers that focus on this topic. I also agree that this perspective can result in other useful results down the line with respect to integration of causality and machine learning research.

On the other hand, I believe that the current paper's contribution to this cause borders on marginal. While the results can be useful for specific applications, in the paper's current state the extent of these applications seem to be limited. These limitations include the lack of an efficient algorithm for t-edge orientation, causal sufficiency assumption, the method inheriting (some) problems associated with cond. independence-based causal discovery methods, and lack of persuasive real applications. Perhaps an improved discussion of potential current applications and experiments on real datasets that indeed has type constraints would help the paper to better justify its arguments. Despite its shortcomings, I recommend the acceptance of the paper, given its potential downstream improvements by the related scientific community.

In addition to my comments above, I list some of the more specific comments and questions I have about the paper:
- How would the authors imagine their work to develop causal discovery in the presence of latent confounders?
- How does authors' theory and proposed algorithm work when we do not know the types of certain variables and/or when there is no constraint between certain pairs of variables in either direction?
- I think the concept t-edge is confusing. t-DAG is an augmented version of the original object (DAG). But t-edge is something completely different than the original object (a graph edge), i.e. while a t-edge is a set of edges, an edge is just a tuple. I would recommend using something similar to a t-edge-set. I imagine the authors think that this allows formulating the relationship between types as a higher-level DAG; but in its current state this terminology is more confusing than helpful.

**Summary:**

The authors demonstrate that if additional assumptions are made on the vertices and the edges of a causal graph the resulting Markov equivalence class becomes much smaller. Their assumptions are: that vertices belong to predefined classes and causal relations can only be in one direction between any two classes.

---

> ### Author Response · Authors · 2021-12-02
> **Response to reviewer Dzzm**
>
> Thank you for your comments. The general response (above) contains answers to your concerns about causal sufficiency (Point 1), the limitations of conditional independence-based methods (Point 2), and an enhanced discussion of potential applications (Point 3). Your other comments are addressed below.
>
> ### 1. Lack of persuasive real applications
>
> Our main objective is to show the benefits of using assumptions on variable types in causal discovery. Such benefits are clearly visible in our simulation studies. We agree that showing results on real-world datasets would have been a plus. Our results on the pseudo-real data are a step in this direction. Due to the inherent complexity of working with real-world datasets (e.g., lack of ground truth graph), we have resorted to surveying the literature for studies where our theoretical framework would have been applicable. In future work, we will strive to develop collaborations with domain experts to apply our work to real-world problems.
>
> ### 2. How does authors' theory and proposed algorithm work when we do not know the types of certain variables and/or when there is no constraint between certain pairs of variables in either direction?
>
> We answer this question in two parts. The first case, where the type of some variables is unknown, is covered by the current version of our theoretical framework. The second case, where type consistency does not apply to some variables, would require a minor extension of our framework, which we have chosen to leave out to avoid complicating the presentation.
>
> **a) When we do not know the types of certain variables.**
> Variables for which the type is unknown should be assigned to a unique type of which they are the only instance. This ensures that the type consistency assumption still constrains their interactions with other types. Alternatively, one could think of assigning all variables with unknown types to a single (common) type. However, this is incorrect, since it forces all such variables to interact with other types in the same direction. We will add a comment about this in the appendix.
>
> **b) When there is no constraint between certain pairs of variables.**
> To handle this case, one would need to add an “exclusion set” to the definition of t-DAGs, which would contain vertices (i.e., variables) to which the type consistency assumption is not applied. We left out this case to avoid complicating the presentation. Note that our Theorem 12 would still hold in this case, but the statement of Corollary 13 would need to be restricted to the case where the exclusion set is empty. We will also add a comment about this in the appendix.
>
> ### 3. The concept t-edge is confusing [...] I would recommend something similar to a t-edge-set.
>
> We understand the reviewer’s concerns and agree that the word t-edge-set would clearly expose the *set* nature of t-edges. However, as mentioned by the reviewer, t-edges are abstractions that allow reasoning about the orientation of type relationships in a higher-level graph. While we expose their set nature in the first place, we quickly move on to an edge-like notation $t_i \xrightarrow{t} t_j$, which we believe facilitates the presentation of our results. We will keep this in mind and try to improve the presentation in the camera-ready.

---

> > ### Comment · Reviewer_Dzzm · 2021-12-17
> > **Thanks for the response**
> >
> > I thank the authors for their further comments and clarifications, and am looking forward to reading the published version of their paper.

---

### Author Response · Authors · 2021-12-02
**General response to all reviewers (1/3)**

We thank the reviewers for their positive feedback and for their insightful questions that have led us to reflect more upon this work and its future extensions. We address questions that were raised by more than one reviewer in this common response and respond to any reviewer-specific concerns separately.

### 1. Extension to semi-Markovian settings (latent confounding) [Reviewers Dzzm, Q9JM]
A full characterization of the benefits of typing assumptions in the presence of latent confounding goes beyond the scope of this work. However, we view this as an important future direction, since an extension of our framework to this case would strengthen its practical applicability. As a specific example of how our typing assumptions could help reduce the size of the equivalence class in the presence of latent confounding, consider a PAG with the edge $a \mathbin{{\circ}\mspace{-2mu}{\rightarrow}} b$, where $\mathbin{{\circ}\mspace{-2mu}{\rightarrow}}$ denotes that some MAGs entailed by the PAG contain either $a \rightarrow b$ or $a \leftrightarrow b$ (confounded) and where $a$ and $b$ are respectively of type $t_1$ and $t_2$. If we know that $t_2 \xrightarrow{t} t_1$ (e.g., inferred from a two-type fork in another part of the graph), then we know that $a \leftrightarrow b$, i.e., the relation between $a$ and $b$ is due to a latent confounder. We will pursue this direction further and explore adaptations of algorithms for this setting (e.g., FCI) as part of future work.

### 2. Other types of structure learning algorithms (other than PC) [Reviewers Dzzm, Q9JM]
Our typing assumptions are broadly applicable and are not limited to variants of the PC algorithm. In this work, we chose to focus on PC due to the simplicity and ubiquity of this algorithm. Nonetheless, t-MEC consistent versions of other algorithms, e.g., score-based ones, are definitely possible to derive. For example, consider the score-based algorithm DCDI [(Brouillard et al. 2020)](https://arxiv.org/abs/2007.01754), which searches for the highest-scoring DAG using continuous optimization. In this algorithm, the adjacency matrix of the graph is parameterized by a matrix of parameters representing the probability of having an edge between each pair of variables. It would be possible to directly enforce type-consistency by adding t-edge orientation probabilities to the parametrization. While such extensions are possible, they go beyond the scope of this paper and we keep them for future work.

---

> ### Author Response · Authors · 2021-12-02
> **General response to all reviewers (2/3)**
>
> ### 3. Important comment about the nature of variable types [Reviewers Nb5f] and potential practical applications [Reviewer Dzzm]
>
> > I do wonder how often practitioners will find themselves in a situation where they can a) confidently partition variables into distinct types and b) have no idea how these types relate to each other. In my experience applying these methods, a) and b) tend to move together, since having a basic temporal ordering of variables is quite common in practice. (Reviewers Nb5f)
>
> We agree that it is very plausible that an expert who would be able to attribute types to variables would also have some intuition about how these types are related. Our framework is compatible with this setting; the orientation of some t-edges may be known a priori. However, our results also work in the case where such prior knowledge is either unavailable or unreliable. Below, we give two examples of cases where an expert may be able to partition variables into types, but where their ordering may be unknown. We will add these examples to the appendix, since these may help practitioners formulate their problems into our proposed framework.
>
> **Example 1 - Time-related types with missing information.**
> Consider the setting where variables are collected over multiple days. This corresponds to a typical tiered background knowledge setting, where we let tiers correspond to types, i.e., the day on which variables were measured (e.g., Day 1 variables, …, Day n variables). Now, consider the case where the exact date of measurement for some types is missing (or unreliable), e.g., for privacy consideration. That is, we know which variables were collected simultaneously, but we don’t know when. While it is not possible to order such types with respect to the others, we know that their inter-type causal relationships follow the arrow of time and that all variables in a type are at the same position in time. In contrast with standard tiered background knowledge, our framework allows for the inclusion of such partial information.
>
>
> **Example 2 - Entities as types.**
> In some use-cases, types could be used to represent distinct entities, each characterized by multiple measured variables. For example, such entities could be devices on a network for which telemetry data is available, users in a social network and their behavioral characteristics, or machines in a manufacturing production line and their input/output quantities (akin to [Marazopoulou et al. (2016)](https://arxiv.org/abs/1605.04056)). In such settings, the nature of the entities is not necessarily sufficient for an expert to have an intuition about their ordering. For example, we may know that two sets of variables are from two distinct network devices, but not how these devices are related within the network topology. When it is reasonable to assume that entities interact in a directional manner (e.g., producer/consumer, influencer/follower relationships), our typing assumptions can be applied by assuming variables of each entity to be of the same type.

---

> > ### Author Response · Authors · 2021-12-02
> > **General response to all reviewers (3/3)**
> >
> > ### 4. Extension: learning the types of variables [Reviewers Q9JM, Nb5f]
> > Thank you both for your comments and suggestions. An extension to the case where variable types are learned from data goes beyond the scope of the current paper. Several details remain to be worked out, but we attempt to give an intuition of what we have in mind below.
> >
> > > I could not tell from the discussion exactly what they had in mind here. I believe it’s important that types in this context are not simply categories: rather, they have a substantive causal flavor. [Reviewer Q9Mj]
> >
> > What we have in mind is a setting in which observational data is complemented with metadata for each variable (e.g., a textual description, a DNA sequence, a picture), which would be used to assign them to types (akin to clustering). We absolutely agree that assigning variables to arbitrary categories, deprived of causal semantics, is not enough. Rather, we plan to explore an approach where variable type assignments are learned in parallel with the graph structure and where being assigned to a specific type limits the possible interactions of a variable according to the type consistency assumption. Achieving this successfully will likely necessitate additional assumptions, e.g., (at the very least) that the metadata contains relevant information. A thorough study of such assumptions is kept for future work.
> >
> > > Is it possible to incorporate some notion of uncertainty about types? The motivation here is that if I am unsure of how types relate to each other, I am probably also less sure of which variables belong to which type. Perhaps the authors may be able to sketch out a sensitivity analysis strategy that addresses this. [Reviewer Nb5F]
> >
> > Identifying the types of variables from metadata is clearly non-trivial and taking uncertainty into account will very likely be necessary. For example, the metadata available for a variable may not be sufficient to assign it to a single class with high certainty. Relying on a sensitivity analysis strategy to measure how much the equivalence class would change if a variable were assigned to one type or another is an interesting idea. For example, if two of the learned types ($t_i$ and $t_j$) correspond to partitions of a true type ($t$), one may learn that assigning a variable with type $t$ to either $t_i$ or $t_j$ has no impact on the resulting equivalence class. We will keep this in mind when further pursuing this idea.

---

### Decision · Program_Chairs · 2022-01-12

**Decision:**

Accept (Oral)

**Comment:**

This paper studies the problem of causal DAG discovery under an additional assumption: variables are typed and between two types of variables, causal relations, if any, are all of the same direction type-wise. The paper presents useful results on the size of an Markov equivalence class given the assumption, and shows empirically how this assumption, if warranted, can significantly improve the informativeness of causal discovery from observational data. These contributions are in my view sufficiently original and significant. Both the clarity and the technical quality are very good, as acknowledged by all reviewers.

A main worry concerns whether there are many real situations in which the typing assumption can be reasonably adopted. Although the authors made plausible arguments for the potential applications of their approach, real applications are yet to be made. The lack of experiments on real datasets is a notable drawback. Other, more minor concerns were raised in the reviewers' reports, to which the authors provided good responses.

Overall I agree with the reviewers that this paper is clearly acceptable. I also judge that it will be of interest to a relatively wide audience of this conference. I therefore recommend an oral presentation.